# A Novel Four Mitochondrial Respiration-Related Signature for Predicting Biochemical Recurrence of Prostate Cancer

**DOI:** 10.3390/jcm12020654

**Published:** 2023-01-13

**Authors:** Zhongyou Xia, Haolin Liu, Shicheng Fan, Hongtao Tu, Yongming Jiang, Hai Wang, Peng Gu, Xiaodong Liu

**Affiliations:** 1Department of Urology, The First Affiliated Hospital of Kunming Medical University, Kunming 650032, China; 2The First Affiliated Hospital of Kunming Medical University, Yunnan Province Clinical Research Center for Chronic Kidney Disease, Kunming 650032, China; 3Department of Urology, Institute of Urology, West China Hospital, Sichuan University, Chengdu 610041, China; 4Department of Urology, The Second Affiliated Hospital of Kunming Medical University, Kunming 650101, China

**Keywords:** biochemical recurrence, prostate cancer, mitochondrial respiratory related gene, prognosis, GSEA

## Abstract

The biochemical recurrence (BCR) of patients with prostate cancer (PCa) after radical prostatectomy is high, and mitochondrial respiration is reported to be associated with the metabolism in PCa development. This study aimed to establish a mitochondrial respiratory gene-based risk model to predict the BCR of PCa. RNA sequencing data of PCa were downloaded from The Cancer Genome Atlas (TCGA) and Gene Expression Omnibus (GEO) databases, and mitochondrial respiratory-related genes (MRGs) were sourced via GeneCards. The differentially expressed mitochondrial respiratory and BCR-related genes (DE-MR-BCRGs) were acquired through overlapping BCR-related differentially expressed genes (BCR-DEGs) and differentially expressed MRGs (DE-MRGs) between PCa samples and controls. Further, univariate Cox, least absolute shrinkage and selection operator (LASSO), and multivariate Cox analyses were performed to construct a DE-MRGs-based risk model. Then, a nomogram was established by analyzing the independent prognostic factor of five clinical features and risk scores. Moreover, Gene Set Enrichment Analysis (GSEA), tumor microenvironment, and drug susceptibility analyses were employed between high- and low-risk groups of PCa patients with BCR. Finally, qRT-PCR was utilized to validate the expression of prognostic genes. We identified 11 DE-MR-BCRGs by overlapping 132 DE-MRGs and 13 BCR-DEGs and constructed a risk model consisting of 4 genes (*APOE*, *DNAH8*, *EME2*, and *KIF5A*). Furthermore, we established an accurate nomogram, including a risk score and a Gleason score, for the BCR prediction of PCa patients. The GSEA result suggested the risk model was related to the PPAR signaling pathway, the cholesterol catabolic process, the organic hydroxy compound biosynthetic process, the small molecule catabolic process, and the steroid catabolic process. Simultaneously, we found six immune cell types relevant to the risk model: resting memory CD4+ T cells, monocytes, resting mast cells, activated memory CD4+ T cells, regulatory T cells (Tregs), and macrophages M2. Moreover, the risk model could affect the IC50 of 12 cancer drugs, including Lapatinib, Bicalutamide, and Embelin. Finally, qRT-PCR showed that *APOE*, *EME2*, and *DNAH8* were highly expressed in PCa, while *KIF5A* was downregulated in PCa. Collectively, a mitochondrial respiratory gene-based nomogram including four genes and one clinical feature was established for BCR prediction in patients with PCa, which could provide novel strategies for further studies.

## 1. Introduction

Prostate cancer (PCa) is the most commonly diagnosed malignant tumor of the male genitourinary system, ranking second and fifth in morbidity and mortality worldwide [1]. According to GLOBOCAN 2020, an estimated 191,930 new patients with PCa were diagnosed globally [2]. Radical prostatectomy (RP) and radical radiotherapy (RT) are the recommended curative treatments for clinically organ-confined PCa, and technological advances have improved the efficacy of both RP and RT. However, approximately 20–60% of patients who receive radical treatment encounter biochemical recurrence (BCR) within 10 years [3]. Biochemical recurrence is when prostate-specific antigen (PSA) levels rise after the treatment of PCa, with certain PSA not reaching a consistent level. Routinely, we define PSA > 0.2 ng/mL after radical surgery or PSA > 2 ng/mL higher than the post-radiation PSA nadir as BCR [4,5]. It is reported that 24–34% of PCa patients with BCR exhibit distant metastasis [6]. Once the patients demonstrate signs of BCR, the treatment becomes very tricky. Antonarakis et al. reported that the median time for metastasis-free survival of patients with BCR was 10 years [7]. Studies have shown that clinicopathological factors, such as clinical staging, PSA, Gleason score, and surgical margin, cannot accurately predict BCR [8]. Therefore, it is urgent to seek new and more accurate BCR predictive markers and explore the pathogenesis of PCa.

Mitochondria are the core site of cellular energy production in the form of adenosine triphosphate (ATP) through oxidative phosphorylation (OXPHOS) of glucose [9]. Recent studies have shown that targeting the energy metabolism of cancer cells might be a new and promising area for selective tumor treatment [10]. Emerging evidence suggests that mitochondria are involved in fatty acid metabolism, reactive oxygen species (ROS) production, and cell apoptosis, which can change cancer cell progression and survival [11,12]. In 1956, Warburg first observed a close relationship between mitochondrial respiratory defects, aerobic glycolysis, and cancer [13]. A recent study has verified that impairment of mitochondrial respiration can inhibit the migration and invasion of breast cancer, mouse melanoma, and PCa cells [14]. Among patients with colorectal cancer, a high mitochondrial respiration level in the tumor samples has been associated with poor survival [10]. Roy et al. showed that the mitochondrial respiration profile might serve as a biomarker for the identification of leukemic cells [15]. Mitochondrial DNA (mtDNA) encodes essential subunits of the mitochondrial respiratory chain that act as energy sources and facilitate tumor proliferation and invasion. Kalsbeek et al. used next-generation sequencing to examine mitochondrial genomes from PCa tissue biopsies, which showed a positive correlation of the total burden of acquired mtDNA variants with the elevated Gleason score at diagnosis and BCR [9]. Furthermore, it was found that mitochondrial damage and genome change were related to tumor growth, metastasis, and BCR [16]. Although studies have established that mitochondrial respiration plays a critical role in tumor development and the anti-tumor process, the relationship between mitochondrial respiration and BCR in PCa is still unclear.

In our present study, we constructed a risk signature based on four mitochondrial respiration-related genes (MRGs) using The Cancer Genome Atlas (TCGA) sequencing data of PCa, and the GSE116918 dataset was used to verify the stability of this model. Furthermore, the correlation of MRGs with clinical features and the tumor immune microenvironment was analyzed. Finally, a nomogram was constructed based on four MRGs and clinicopathological factors which demonstrated good performance for predicting prognosis in patients with PCa.

## 2. Materials and Methods

### 2.1. Data Sources

The expression profile and clinical information of 499 PCa samples and 52 controls were obtained from The Cancer Genome Atlas Prostate Adenocarcinoma (TCGA-PRAD) via the University of California Santa Cruz Xena (UCSC Xena) database (https://xenabrowser.net/, accessed on 22 June 2022). Of the 499 PCa samples, 435 had BCR information (342 non-BCR samples and 93 BCR samples), of which 400 were accompanied by a specific BCR time, and 315 of the 400 PCa samples had survival information. The external validation dataset, GSE116918, which included 248 localized/locally advanced patients with PCa commencing radical radiotherapy (with ADT) (192 non-BCR samples and 56 BCR samples), was downloaded from the GEO database (https://www.ncbi.nlm.nih.gov/geo/, accessed on 22 June 2022). Additionally, 785 MRGs were sourced via GeneCards (https://www.genecards.org/, accessed on 24 June 2022). The clinical characteristics of patients with PCa in the TCGA-PRAD cohort and GSE116918 dataset are shown in Appendix A. The GSE70768 dataset (consisting of 113 PCa and 73 normal samples) and the GSE55945 dataset (consisting of 13 PCa and 8 normal samples) were applied to validate the expression of the model genes.

### 2.2. Identification of Differentially Expressed Mitochondrial Respiratory BCR-Related Genes (DE-MR-BCRGs) and Enrichment Analysis

Initially, overlapping genes of 785 MRGs with 499 PCa samples and 52 controls from TCGA-PRAD were selected. Then, differential analysis was performed on the overlapped genes to identify differentially expressed MRGs (DE-MRGs) between PCa samples and controls with the thresholds of *p* < 0.05 and |log (FC)| > 0.5 by Limma (version 3.44.3) [17]. Simultaneously, the same differential analysis was applied to screen BCR-related differentially expressed genes (BCR-DEGs) between 342 non-BCR and 93 BCR samples from TCGA-PRAD. Subsequently, DE-MRGs were overlapped with BCR-DEGs through jvenn (http://jvenn.toulouse.inra.fr/app/example.htm, accessed on 8 July 2022), and the intersected genes were considered as DE-MR-BCRGs. The clusterProfiler (version 3.8.1) [18] was used to perform functional annotation and pathway enrichment analyses, including Gene Ontology (GO) and Kyoto Encyclopedia of Genes and Genomes (KEGG) pathway analysis, for DE-MR-BCRGs. A *p*-value < 0.05 was considered statistically significant.

### 2.3. Construction and Validation of the BCR-Based Prognostic Risk Score Model

Univariate Cox regression analysis was initially employed on the DE-MR-BCRGs; genes with a *p*-value of < 0.05 were further screened by least absolute shrinkage and selection operator (LASSO) regression. LASSO was performed by glmnet (version 4.1-1), with the set of family = Cox and nfold = 20. The genes screened by LASSO when λ was minimum were optimized stepwise to build a multivariate Cox BCR-based prognostic risk score model, and the genes in the model were deemed as model genes.

The prognostic risk score model was constructed with the formula: risk score $=\overset{n}{\underset{n=1}{\sum }}coef_{i}*x_{i}$. The prognostic performance of the risk score model was evaluated by time-dependent receiver operating characteristic (ROC) curve analysis within 1, 3, and 5 years with “survivalROC” (version 1.16.1) [19]. Next, prognostic risk scores were generated for the 400 PCa samples with a specific BCR time in TCGA-PARD. The 400 PCa samples were divided into high-risk and low-risk groups according to the median value of their risk score. Then, a Kaplan–Meier (K–M) survival curve was constructed for each risk group to compare their survival differences using the surv_cutpoint function. Moreover, a risk curve and a scatter plot of the risk groups, and a heatmap of model genes in each risk group were generated. Additionally, the same evaluation procedures were employed in the external validation set (GSE116918) to further verify the applicability of the risk model.

### 2.4. Establishment of Nomogram

In order to explore the correlation between clinicopathological features and risk score grouping, the data for PSA (≥10, <10), race (white, others), Gleason score (≥7, <7), age (≥60, <60), and clinical stage (≥T3, <T3) were extracted from the 400 patients with PCa in TCGA-PARD and the correlations with risk score grouping were determined using Pearson correlation. Similarly, the correlation between clinicopathological features and risk score grouping was also analyzed in the training set (GSE116918). However, since there was no information on the race of the patients, only the PSA, Gleason score, age, and clinical stage of the 248 PCa samples were correlated and analyzed with risk score grouping. Next, the correlation between the risk scores and the five clinicopathological features was analyzed; the clinicopathological features with statistical significance (*p* < 0.05) and risk score of the 315 PCa samples with survival information in TCGA-PARD were introduced into univariate and multivariate Cox proportional hazards regression analyses. The results with *p* < 0.05 in both Cox analyses were considered independent prognostic factors to construct an independent prognostic model. Furthermore, the model was evaluated and validated by plotting its ROC curves of the training and validation sets, respectively. In addition, the independent prognostic model was visualized through rms (version 6.2-0) [20] into a nomogram, and its prognostic performance was evaluated by Harrell’s concordance index (C-index) and the slopes of its calibration curves.

### 2.5. Correlations between Risk Score and Enriched Pathways and Terms

The ssGSEA algorithm scored the functional enrichment results to investigate the molecular mechanisms of model genes. Subsequently, to detect the pathways significantly associated with risk scores, the correlation between each ssGSEA and the risk scores was computed using the Pearson algorithm.

### 2.6. Tumor Microenvironment (TME) Analyses

To detect the differences between the infiltration of the two risk groups, CIBERSORT was applied to calculate the proportion of various immune cell types from the LM22 gene set in each sample from both risk groups. The proportion of each immune cell was further compared between high- and low-risk groups. Moreover, Pearson correlation analysis was used to analyze the correlations between 22 immune cells, model genes, and independent prognostic factors.

### 2.7. Susceptibility Analysis of Immune and Chemotherapeutic Drugs

The Tumor Immune Dysfunction and Exclusion (TIDE) score of each risk group was calculated. A higher TIDE score indicates that tumor cells are more likely to induce immune escape, thus revealing a lower response rate to ICI therapy. The rank-sum test was employed to calculate the difference in TIDE score between the low-risk and high-risk groups in the training set. Pearson analysis was used to analyze the correlation between risk and TIDE scores. Finally, a ridge regression model was constructed to predict the IC50 of the 138 cancer drugs in pRRophetic by the pRRophetic algorithm (version 0.5) [21], according to cell line expression profiles in GDSC and gene expression profiles in TCGA. In the training set, the Spearman correlation analysis was used to evaluate the association between risk score and drug IC50. The IC50 of drugs that satisfied |correlation coefficient| > 0.3 and *p* < 0.005 were visualized.

### 2.8. Expressions Validation of Model Genes

A qRT-PCR assay was used to validate the accuracy of the expression results of model genes on 12 culture bottles with a density of 70–90%, containing 3 independent cultures each of RWPE-1 (normal human epithelial prostate cells), PC-3, LNcap, and DU145 (PCa cells). Total RNA for the qRT-PCR was extracted using TRIZol (Thermo Fisher, Shanghai, China), mRNA was reverse transcribed into cDNA, and qPCR reactions were performed using the SureScript-First-strand-cDNA-synthesis-kit (Servicebio, Wuhan, China) following the manufacturer’s protocol. The qRT-PCR reaction consisted of 3 µL of cDNA, 5 µL of 2xUniversal Blue SYBR Green qPCR Master Mix (Servicebio, Wuhan, China), and 1 µL each of forward and reverse primers. PCR was performed in a BIO-RAD CFX96 Touch TM PCR detection system (Bio-Rad Laboratories, Inc., Hercules, CA, USA) under the following thermal cycling conditions: 40 cycles at 95 °C for 60 s, 95 °C for 20 s, 55°C for 20 s, and 72 °C for 30 s with specific PCR primers shown in Appendix A. With GAPDH as the internal control, the 2^−△△Ct^ method was used to compute gene expressions. GraphPad Prism 5 was used to plot and calculate the statistical significance.

### 2.9. Statistical Analysis

All statistical analyses based on public datasets in this study were performed in R (Version 4.2.0), while the experimental data were analyzed in GraphPad Prism 5. Package ggplot2 was used to plot box plot and Vacano plot, and package Pheatmap was applied to plot heat map. The Wilcox test and Student’s *t*-test were performed to analyze two groups. A log-rank test was conducted to compare the OS differences between two risk groups.

## 3. Results

### 3.1. Identification and Enrichment Analysis of DE-MR-BCRGs

We identified 132 DE-MRGs (43 upregulated, 89 downregulated) and 13 BCR-DEGs (10 upregulated, 3 down-regulated) through differential analysis (Appendix A). Subsequently, we obtained 11 DE-MR-BCRGs by overlap analysis, which included APOE, CDK1, CPT1B, CXCL8, CYP27A1, DNAH8, EME2, FOXH1, KIF5A, MTFR2, and PAH (Figure 1A). The DE-MR-BCRGs were enriched in 8 GO terms and 11 KEGG pathways based on the selection threshold of *p* < 0.05. For instance, these genes were primarily involved in five biological processes such as the cholesterol catabolic process, the sterol catabolic process, the small molecule catabolic process, the steroid catabolic process, and the organic hydroxy compound biosynthetic process, and three molecular functions, including microtubule motor activity, monooxygenase activity, and cytoskeletal motor activity (Figure 1B). The dot chart represents the top 10 enriched pathways in terms of the KEGG pathways, which mainly included cholesterol metabolism, signaling pathway, alcoholic liver disease, Salmonella infection, and cellular senescence (Figure 1C).

### 3.2. A BCR-Based Prognostic Risk Score Model of 4 Genes Was Constructed

The univariate Cox regression analysis led to the selection of nine DE-MR-BCRGs for LASSO analysis (Table 1). For the accuracy of the model, lambda.min (0.01027) was selected as λ for model construction, and eight genes were screened out, including *APOE*, *CDK1*, *CPT1B*, *CYP27A1*, *DNAH8*, *EME2*, *FOXH1*, *KIF5A*, and *MTFR2* (Figure 2A). Finally, four model genes were selected through multivariate Cox analysis: *APOE*, *DNAH8*, *EME2*, and *KIF5A* (Figure 2B). Subsequently, to evaluate the prognostic value of the risk model, the expression levels of the four model genes were obtained in the TCGA dataset, and the risk score = APOE expression × 0.294748712013631 + DNAH8 expression × 0.0715331616294412 + EME2 expression × 0.48193526167778 + KIF5A expression × 0.158063686901454. From the ROC curves of the model, we found that the AUCs of all time periods were greater than 0.7, indicating the effectiveness of the prognostic risk score model (Figure 2C). Furthermore, 400 PCa samples were divided into high-risk (117 samples) and low-risk groups (283 samples), according to the median value of the risk scores. The K–M curves of the risk groups revealed that the survival probability of the low-risk group was significantly higher (Figure 2D). When plotting the risk curve, we noticed that the extreme value of the risk score was large (26.83), so all the risk scores exceeding the value of 10 were depicted as a risk score of 10. Moreover, the risk curves illustrated that the high-risk group had a higher risk score than the low-risk group, and samples in the high-risk group tended to be BRC samples (Figure 2E). Furthermore, the heatmap also showed that the model genes were upregulated in the high-risk group (Figure 2F).

As the number of BCR samples within 1 and 2 years in the GSE116918 dataset was very small (only 2 and 9 cases), accounting for 3.6% of the total sample, we generated the ROC curves for 3 and 5 years with the external validation set. Moreover, the 248 PCa samples were separated into high-risk (49 samples) and low-risk groups (199 samples). Consistent with the training set, the validation results demonstrated an AUC of a 3-year ROC curve greater than 0.6, indicating that the risk model could effectively predict the prognosis model (Appendix A).

### 3.3. Construction of A Nomogram

Among the five clinicopathological features, we found that the risk score was significantly correlated with the Gleason score and clinical stage in the training set. Meanwhile, we found that only the Gleason score and risk score were significantly correlated in the validation set (Table 2 and Table 3). Moreover, the risk score comparison between subgroups in clinicopathological features illustrated that the risk score was significantly higher in samples with Gleason score ≥ 7, age ≥60, and clinical stage ≥ T3 (Figure 3A). In order to obtain the independent prognostic factors of PCa patients with BCR, the Gleason score, age, clinical stage, and risk score of 315 PCa samples in TCGA-PARD were introduced into univariate and multivariate Cox analyses. The results demonstrated that the *p*-values of the risk score and Gleason score were less than 0.05 in both Cox analyses. Therefore, the risk score and Gleason score were used to establish the independent prognostic model (Table 4, Figure 3B). The AUC of the ROC curves for the independent prognostic model was 0.813, 0.827, and 0.789 at 1-, 3-, and 5-year time nodes in the training set (Figure 3C). Moreover, the effectiveness of the independent prognostic model was higher than the risk model at 3- and 5-year time nodes. The AUC of the 5-year ROC curve in the validation set was found to be 0.53, revealing that the effectiveness also increased compared with the risk curve (Figure 3D). Finally, a nomogram of the independent prognostic model was constructed with C-index = 0.7295, indicating that the prediction of the nomogram was accurate (Figure 3E). Compared to age (AUC = 0.555), the factors, such as the Gleason score (AUC = 0.739) and PSA (AUC = 0.809), showed a striking prognostic predictive efficiency for overall survival (OS) rates in the TCGA dataset (Appendix A). The risk score (AUC = 0.808) showed an excellent prognostic predictive efficiency for OS rates in the TCGA dataset (Appendix A). In addition, the slopes of the 1-, 3-, and 5-year nomogram calibration curves were close to 1, which also meant that the nomogram could effectively predict the prognosis of PCa patients with BCR (Figure 3F).

### 3.4. Correlations between Risk Score and Enriched Pathways and Terms

The Pearson correlation analysis showed three significant correlations between KEGG pathways and the risk model. The risk model was positively correlated with Alzheimer’s disease but negatively correlated with the PPAR signaling pathway and primary bile acid biosynthesis. Furthermore, we found four correlations of the GO terms with the risk model, all of which were negative, including the cholesterol catabolic process, the organic hydroxy compound biosynthetic pathway, the small molecule catabolic process, and the steroid catabolic process (Table 5). The results suggested that the four model genes (*APOE*, *DNAH8*, *EME2*, and *KIF5A*) may influence the prognosis and progression of PCa by regulating these signaling pathways.

### 3.5. Comparison of TME between the Risk Groups

Analysis by CIBERSORT revealed that the proportion of six immune cell types was significantly different between the two risk groups. For instance, the proportions of resting memory CD4+T cells, monocytes, and resting mast cells were higher in the low-risk group. In contrast, in the high-risk group, activated memory CD4+ T cells, regulatory T cells (Tregs), and M2 macrophages were higher (Figure 4A,B). Furthermore, it can be observed that among the four model genes, the Gleason score, and the risk score, the strongest positive correlation existed between *APOE* and M2 macrophages, and the strongest negative correlation was found between *APOE* and resting memory CD4+ T cells (Figure 4C). We also generated the correlation scatter plots for these two correlations (Figure 4D). Subsequently, violin plots comparing the expression between the two risk groups showed that *APOE* and M2 macrophages were higher in the high-risk group, while the resting memory CD4+ T cells showed the opposite trend (Figure 4E).

### 3.6. Susceptibility Analysis of Immune and Chemotherapeutic Drugs

We next compared the TIDE score between the two risk groups to detect the response to ICI. The TIDE score was higher in the low-risk group and was negatively correlated with the risk score (Figure 5A,B). Moreover, we found positive correlations between the risk score of the training set and the IC50 of 12 drugs (Lapatinib, Bicalutamide, Embelin, Erlotinib, Bexarotene, A.770041, Z.LLNle.CHO, FH535, Imatinib, Cyclopamine, AZD8055, and MG.132), indicating that PCa patients with high risk scores could be more resistant to the 12 administered chemotherapies (Figure 5C). Resistance to chemotherapy may be due to the poor prognosis of PCa patients with high risk scores. Moreover, the IC50 of JNK.Inhibitor.VIII and ABT.888 were negatively correlated with the risk scores, indicating that they could benefit patients with high DE-MRGs-based risk scores (Figure 5C).

### 3.7. Validation of the Expression of Four Model Genes by GSE70768 and GSE55945 Datasets

The expression levels of the four model genes were validated in the GSE70768 and GSE55945 datasets, and the results indicated that *APOE*, *DNAH8*, and *EME2* were significantly lower expressed in PCa samples compared to normal samples, while the expression levels of *KIF5A* showed no significance between normal and PCa samples (Figure 6).

### 3.8. Validating the Expression of the Four Model Genes by qRT-PCR

The expression validation of the four model genes was performed on normal human prostate epithelial cells and PCa cells by qRT-PCR. We found a significantly higher expression of *APOE* and *EME2* in PCa cell line groups compared with RWPE-1. Moreover, *DNAH8* expressed significantly higher in PC-3 and DU145 than in RWPE-1, while the expression of DNAH8 was not detected in LNcap lines. However, the expression of *KIF5A* was distinctly higher in the normal prostate epithelial cell group than in all 3 PCa tissue sample groups, which was contrary to the TCGA result (Table 6, Figure 7). Therefore, *APOE*, *EME2,* and *DNAH8* could be considered reliable and precise model genes for PCa.

## 4. Discussion

The occurrence of BCR in patients with PCa after radical treatment indicates the likelihood of distant metastasis and the development of castration-resistant prostate cancer (CRPC) [8]. Although the advancement in salvage treatment regimens, including radiation therapy, androgen deprivation therapy, chemotherapy, and even intensive multimodal therapy, has improved the prognosis of patients with PCa, most still die within 2 to 4 years [22,23]. PSA and the Gleason score are well-known indicators used to predict BCR in PCa patients and grade the risk of BCR after clinical treatment [24]. However, PCa is a highly heterogeneous disease, and determining the prognosis of certain patients could be challenging. Therefore, better and more accurate prognostic indicators are needed to avoid unnecessary over-medical treatment to identify high-risk patients with BCR and guide individual clinical treatment.

With the development of bioinformatics, studies have reported several prognostic models based on gene signatures to predict BCR. Signatures based on multiple gene expressions, including metabolic [25], immune-associated [26], and ferroptosis-related genes [27], were highly associated with the BCR of patients with PCa. However, too many genes in the above signatures limit their clinical application. There is still a lack of a precision molecular targeting index to effectively predict BCR in patients with PCa. In recent years, the alterations of mitochondrial metabolism in the tumor have been the focus of our research. Mitochondria are highly evolved intracellular organelles that control cell energy production, signaling transduction, and cell death [28]. Many core metabolic pathways in the mitochondria, including those of amino acids, lipids, and carbohydrates, as well as oxidative phosphorylation (OXPHOS), are essential for cancer cell proliferation [29]. The rapid proliferation of cancer requires metabolic adaptations to meet the increasing energy demand and to cope with the oxygen-deprived microenvironment [30]. As essential intermediates produced by mitochondria, NADH, NADPH, and FADH2 fuel the electron transport chain (ETC) and OXPHOS to produce energy [11]. One well-recognized strategy is to shift the metabolic flow from OXPHOS or respiration in the mitochondria to glycolysis in the cytosol, also known as the Warburg effect. However, some cancers do not follow Warburg’s rule. Studies have found that OXPHOS in the ETC provides the major sources of energy to promote cancer proliferation, such as colon cancer [31], PCa [30], and chronic lymphocytic leukemia (CLL) [15]. Thus, the ETC may act as a potential therapeutic target for cancer. Previous research has shown that metastatic cancer maintains high rates of O_2_ consumption compared with normal tissues and stimulates mitochondrial biogenesis [31,32]. Rebane-Klemm et al. [29] used high-resolution respirometry to observe mitochondrial respiration in 48 patients with mutated *KRAS* and *BRAF* in colorectal cancer (CRC). The results indicated that CRC patients have a higher level of mitochondrial respiration with poor survival. Furthermore, Roy et al. (15) found that the zeta-chain-associated protein of 70 kD (ZAP-70), a mitochondrial respiration-related prognostic marker, predicted increased maximal respiration in patients with CLL and increased sensitivity of ZAP-70+ cells to Ibrutinib treatment. Mutations in oncogenes, tumor suppressor genes (including TP53 and bcl-2), and mtDNA variation could directly affect mitochondrial respiration and metabolism in PCa [33,34].

Additionally, zinc ion plays a vital role in the energy metabolism of prostate epithelial cells. Studies also showed that a decrease in zinc concentration could power OXPHOS in the ETC during the early development of PCa [35]. Mitochondrial metabolism-related enzymes, such as SUMO-deficient hexokinase 2, bound to mitochondria, could reduce mitochondrial respiration and result in cancer cell proliferation [36]. The previous literature also reported that high expression of OXPHOS-related sulfite oxidase was associated with post-operative BCR in patients with PCa, possibly by inducing PCa cell proliferation [37]. Together, the above findings indicate that mitochondrial respiration is cross-linked with cancer occurrence, development, and recurrence. Hence, we constructed an MRG prognostic model to predict the BCR in patients with PCa.

Our study used machine learning algorithms (univariate and LASSO) to identify the prognostic signatures associated with mitochondrial respiration, consisting of *APOE*, *DNAH8*, *EME2*, and *KIF5A*, which have valuable and independent significance for predicting BCR. Apolipoprotein E (*APOE*), including E2, E3, and E4 isoforms, have pivotal roles in mediating cholesterol and lipid uptake by cells [38]. In addition, *APOE* is also involved in carcinogenesis since it can modulate angiogenesis, cell growth, and metastasis in tumors [39]. Genetic polymorphisms of *APOE* have been reported to influence the growth and progression of many cancers, including colon cancer [39], breast carcinoma [40], and primary brain tumor [41]. High plasma cholesterol concentration was observed in patients with PCa, and APOE mRNA was highly expressed in PCa cell lines and prostatectomy specimens [42,43]. Further studies found that different APOE isotypes are associated with varying aggressiveness of PCa cells, such as non-aggressive PCa cell lines carry the E3/E4 isotypes while aggressive ones carry the E2/E4 isotypes [44]. Utermann et al. [45] observed that the frequency of homozygosity for the *APOE* ε4 allele was increased in PCa compared with normal tissues. Furthermore, Yencilek et al. [38] demonstrated that the *APOE* E3/E3 genotype might be a potential risk factor for PCa and high Gleason scoring.

Genomic variations of dynein axonemal heavy chain (*DNAH*) family members have been frequently reported in multitudes of malignant tumors. A variant of the *DNAH11* gene, rs2285947, is a potential risk factor for ovarian and breast cancer progression [46]. In addition, gene mutations in *DNAH* increase the sensitivity of patients with gastric cancer to chemotherapy [47]. According to a genome-wide RNAi screen, Wang et al. [48] have found that the high expression of *DNAH8* contributes to a greater risk of relapse and poor survival after prostatectomy, possibly by activating the androgen receptor signaling pathway.

*EME2* can restart a stalled fork and regulate the homologous recombination repair pathway [49,50]. In the present study, we found that *EME2* may regulate mitochondrial respiration and affect the BCR in patients with PCa. Therefore, *EME2* could be a potential therapeutic target for PCa. Further, *KIF5A* is a member of the kinesin family, which can modulate the cell cycle, proliferation, and differentiation [51]. Many studies have demonstrated that high expression of *KIF5A* is associated with cancer progression and a poor prognosis, such as in bladder, lung, and breast cancers [52,53,54]. In addition, exome sequencing for 64 tumor samples from 55 PCa patients demonstrated that the *KIF5A* mutation was related to aggressive diseases [55]. Notably, our qRT-PCR result showed that the expression of *KIF5A* is downregulated in PCa cells compared to normal prostate epithelial cells, which is contrary to the result of the TCGA database and may be related to tumor heterogeneity. Finally, we established a nomogram using these signatures and clinical data and evaluated its performance to facilitate clinical decision-making. Through the above analysis, we could suggest that these four MRGs might serve as potential novel target genes for PCa treatment.

Additionally, GO and KEGG analysis revealed that the risk model was closely related to the Peroxisome proliferator-activated receptor (PPAR) signaling pathway, primary bile acid biosynthesis, the cholesterol catabolic process, the organic hydroxy compound biosynthetic process, the small molecule catabolic process, and the steroid catabolic process. Peroxisome proliferator-activated receptors (PPARs) are nuclear transcription factors that play a vital role in regulating growth and differentiation within normal prostate and PCa cells [56]. The activation of the FABP12/PPARγ pathway induces epithelial-to-mesenchymal transition and lipid-derived energy production to promote PCa metastasis [57]. Olokpa et al. [58] found that reduced androgen receptor function could increase the expression of PPARγ and the anti-tumor effects of PPARγ agonists in PCa. In addition, PPARγ derived PCa growth and metastasis by upregulating AKT3 could increase mitochondrial biogenesis levels [59]. Studies have found that the amount of cholesterol is higher in PCa cells compared to normal cells, influencing cancer development and progression [60]. Henrich et al. [61] revealed that reducing cholesterol in bone marrow myeloid cells can render the transduction of PCa extracellular vesicle signaling, thus hindering the bone metastasis of PCa. It was reported that sex steroid hormones, especially androgens (testosterone and dihydrotestosterone), contribute to the growth and progression of PCa [62]. Ahlering et al. [63] demonstrated that testosterone replacement therapy after radical prostatectomy significantly reduced BCR in patients with PCa and delayed the time to BCR. Furthermore, a new steroid compound (steroid-based copper transporter 1 inhibitors) has also been discovered, which can suppress PCa cell proliferation and tumor growth by reducing copper uptake and may act as a novel anti-cancer drug for PCa [64]. However, further research is needed to understand its mechanisms in PCa better.

Activating the immune response to treat cancer has become the cornerstone of modern oncology therapy. Emerging studies have explored the roles of immune cells in PCa [65,66]. Here we showed that Tregs were the most significantly enriched immune cell in the high-risk group. In addition, activated memory CD4+ T cells and M2 macrophage cells were higher in the high-risk group than in the low-risk group. Various immune cells, namely, CD4+ T cells, CD8+ T cells, natural killer (NK) cells, and macrophages, are enriched in the prostate tumor microenvironment [67]. Research showed that the reduction of T cells was correlated with BCR and poor survival in patients with PCa [68,69], which is partly consistent with our results.

Tregs, distinguished by specific markers (CD25, CD4, CD127, and FOXP3), play a vital role in maintaining immune homeostasis. It is reported that Tregs are significantly enriched in PCa tissues and associated with the progression of cancer cells [70]. Vidotto et al. [71] observed that increased FoxP3+ Tregs were associated with PTEN deficiency and lymph node metastasis in patients with PCa. Several possible mechanisms are involved in cancer progression, such as Tregs inhibiting T lymphocytes’ function, NK cells, DCs, and macrophages, or weakening the immune response by secreting immunosuppressive cytokines such as TGF-β and interleukin-10 (IL-10) [67]. Previously, Hu et al. [72] reported that CD4+ T cells contribute to PCa immune evasion and progression. Their team further discovered that infiltrating CD4+ T cells could promote PCa chemotherapy resistance by modulating the CCL5/STAT3 signaling pathway [73]. Due to nutritional deficiencies caused by energy competition between tumor cells and immune cells, some immune cells use lactic acid as an energy substrate. However, studies found that lactate resulting from stromal metabolic reprogramming could modulate CD4+T cell polarization and induces immunosuppressive behavior to promote PCa progression [74]. M2 macrophages are a class of differentiated tumor-associated macrophages (TAMs) associated with poor clinical outcomes in several cancers [75,76]. A recent study found a significant correlation between M2 macrophages and Tregs; M2 macrophages can stimulate lymphocytes to develop into Tregs to promote an immunosuppressive environment in aggressive PCa [77]. Meanwhile, we further analyzed the correlation between immune cell infiltration and biomarkers and found that *APOE* had a significant positive correlation with M2 macrophages and a negative correlation with resting memory CD4+T cells. In anti-atherogenic, *APOE* can induce macrophage conversion from M1 to M2 [78]. Furthermore, Zheng et al. [76] demonstrated that M2 macrophages could transfer functional *APOE* exosomes to neighboring gastric cancer (GC) cells and activate the PI3K-Akt signaling pathway to promote GC migration. Therefore, *APOE* could become the potential target gene of immunotherapy for PCa.

Tumor Immune Dysfunction and Exclusion (TIDE) is a method to predict the immune checkpoint blockade response using gene expression profiles. Patients with cancer and higher TIDE scores could undergo anti-tumor immune escape [79]. Compared to PA-L1 and tumor mutation burden (TMB) indicators, the TIDE score is more accurate in predicting the survival outcome of patients who received immune checkpoint blockade treatment [80]. We observed that the TIDE score was significantly lower in the high-risk group than in the low-risk group, and the finding suggested that patients with PCa are more sensitive to ICB treatment. Moreover, we also evaluated the association between the risk score and the IC50 of the cancer drugs. Our results indicated that PCa patients with high risk scores could be more resistant to the 12 administered chemotherapies (Lapatinib, Bicalutamide, Embelin, Erlotinib, Bexarotene, A.770041, Z.LLNle.CHO, FH535, Imatinib, Cyclopamine, AZD8055, and MG.132). Resistance to chemotherapy may be due to the poor prognosis of patients with PCa. The IC50 of JNK.Inhibitor.VIII and ABT.888 were negatively correlated with the risk score, indicating that these drugs could benefit patients with high DE-MRGs-based risk scores. However, large samples of randomized controlled trials are needed to further validate the effectiveness of the two drugs (NK.Inhibitor.VIII and ABT.888).

Although our study has achieved encouraging results, there are still some limitations. Firstly, this is a retrospective analysis, and selection bias may exist in this study. Secondly, the clinical information of some patients with PCa from the GEO dataset was incomplete. Thirdly, although we performed a multi-faceted, multi-database validation, the amount of data in this study was relatively small, and therefore, the analysis may be biased. Finally, although qRT-PCR has been used to detect the expression of the four mitochondrial respiration-related genes, further experiments in vitro and in vivo are needed to explore the underlying mechanism behind the risk scores and BCR in PCa.

## 5. Conclusions

This study established a mitochondrial respiratory gene-based nomogram including four genes and one clinical feature for BCR prediction in patients with PCa, which could provide novel research references for further studies.

## Figures and Tables

**Figure 1 jcm-12-00654-f001:**
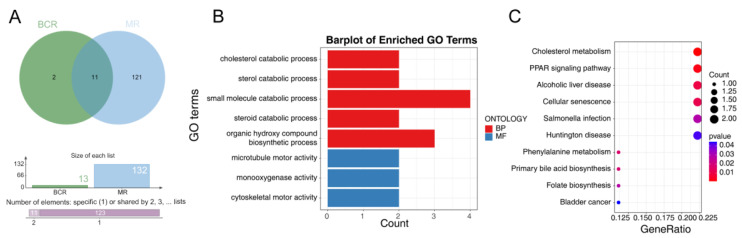
Identification and enrichment analysis of the DE-MR-BCRGs. (**A**) Venn Diagrams of DE-MRGs and BCR-DEGs. (**B**) GO enrichment analysis. (**C**) KEGG enrichment analysis.

**Figure 2 jcm-12-00654-f002:**
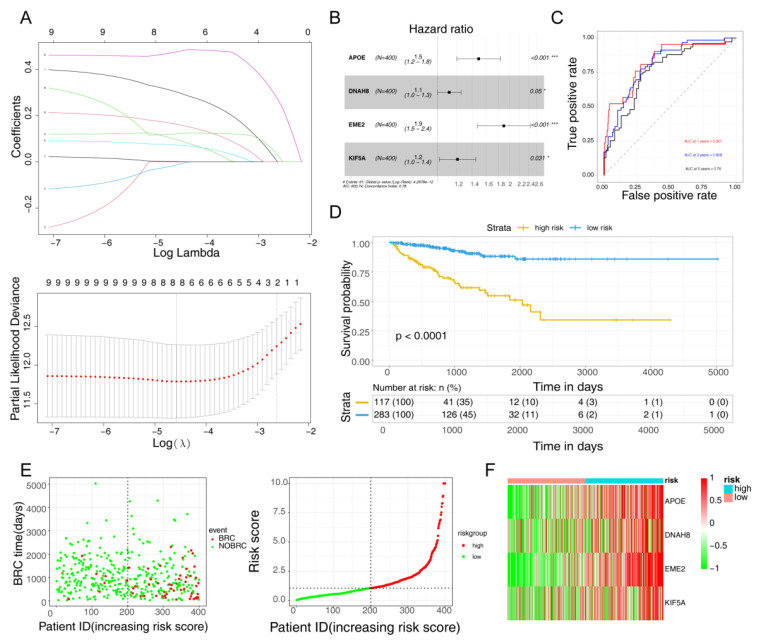
Evaluation of the risk model with risk score in the TCGA dataset. (**A**) LASSO regression analysis. (**B**) Multivariate Cox analysis of model genes. * *p* < 0.05, *** *p* < 0.001. (**C**) ROC analysis of the prognostic risk score model. (**D**) K–M analysis of the prognostic risk score model. (**E**) Scatter plot for BCR and distribution of risk score in the TCGA dataset. (**F**) Heatmap of model genes expression in the TCGA dataset.

**Figure 3 jcm-12-00654-f003:**
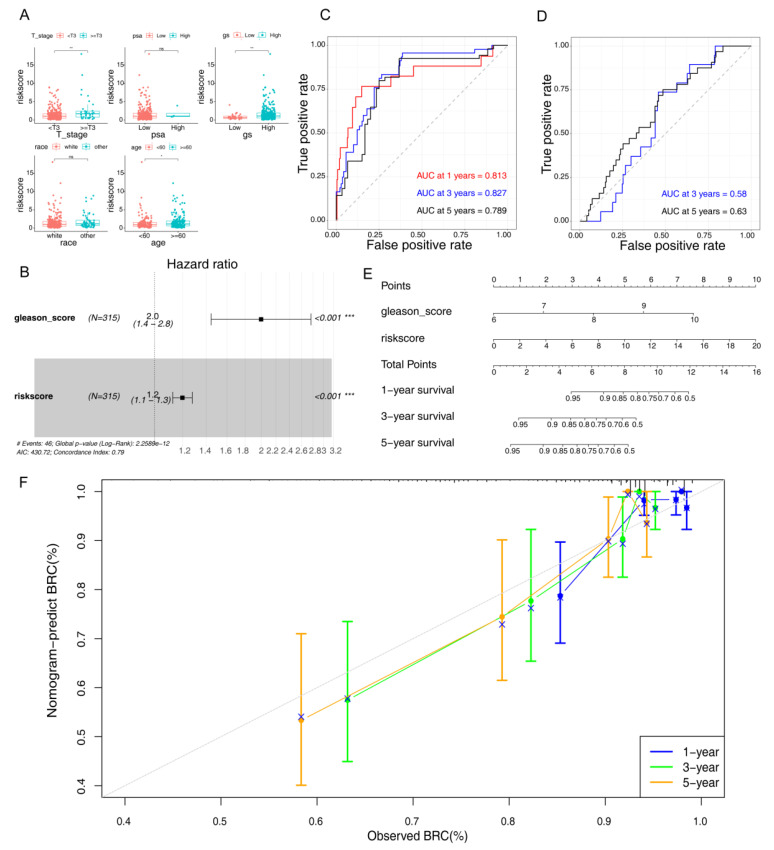
Correlation analysis between the risk model and clinicopathologic features. (**A**) Correlation between risk score and clinicopathologic features. The abscissa represents clinicopathologic features, and the ordinate represents the risk score. * *p* < 0.05; ** *p* < 0.01; ns, not significant. (**B**) Forest plots of univariate and multivariate Cox analyses. *** *p* < 0.001. (**C**) The ROC analysis of the independent prognostic model in the training set. (**D**) The ROC analysis of the independent prognostic model in the validation set. (**E**) A nomogram of the independent prognostic model. (**F**) Calibration curves of the nomogram.

**Figure 4 jcm-12-00654-f004:**
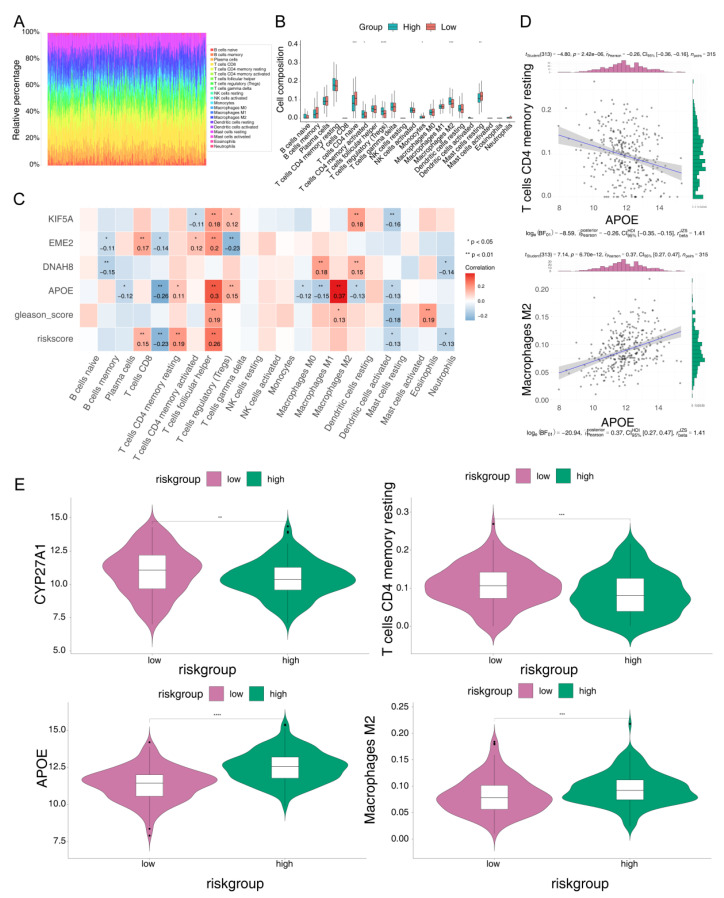
Correlation between risk scores and immune cell infiltration. (**A**) The relative percentage of the 22 immune cell types. (**B**) Box plot of the immune cell proportion. * *p* < 0.05; ** *p* < 0.01; *** *p* < 0.001; **** *p* < 0.0001; ns-not significant. (**C**) Correlation analysis of model genes and immune cells. (**D**) Correlation scatter plots of model genes and immune cells. (**E**) Differences in model genes and immune cells between the high- and low-risk groups.

**Figure 5 jcm-12-00654-f005:**
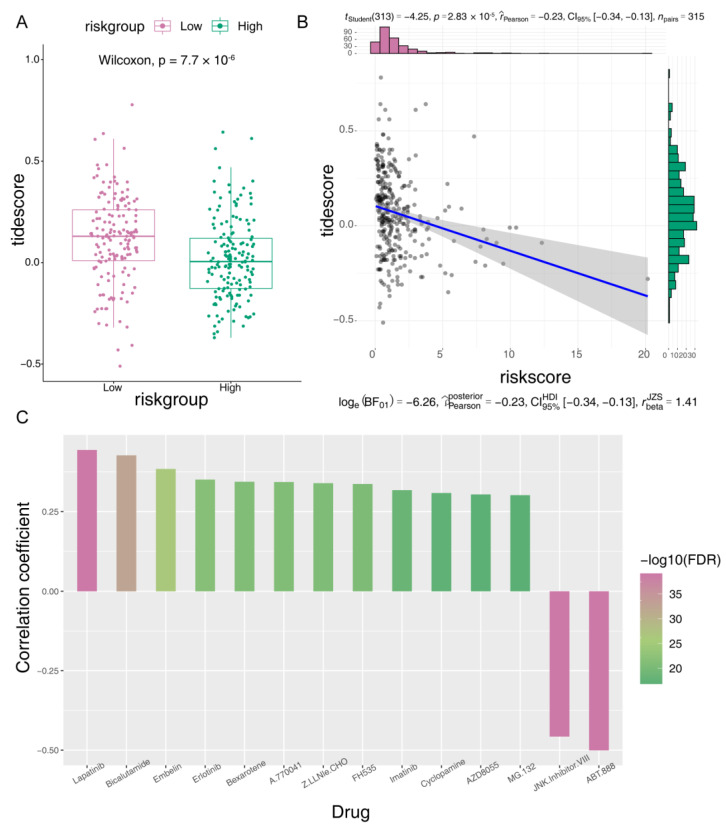
Prediction for immune and chemotherapeutic drugs susceptibility. (**A**) Differences in the TIDE score between the high- and low-risk groups. (**B**) Correlation scatter plots of TIDE score and risk score. (**C**) Correlation analysis of the IC50 of 14 drugs and risk score.

**Figure 6 jcm-12-00654-f006:**
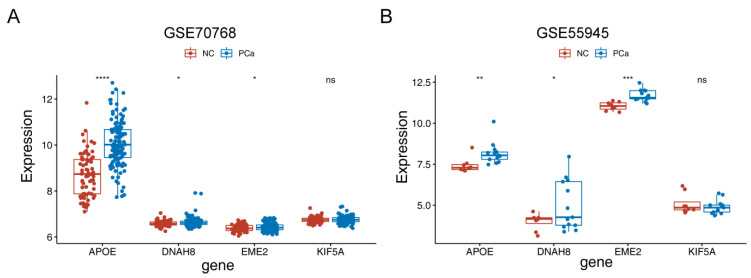
Validation the expression levels of four model genes in the GSE70768 and GSE55945 datasets. (**A**) The expression levels of the four model genes in the GSE70768 dataset. (**B**) The expression levels of the four model genes in the GSE55945 dataset. * *p* < 0.05, ** *p* < 0.01, *** *p* < 0.001, **** *p* < 0.0001, ns represents no significant difference.

**Figure 7 jcm-12-00654-f007:**
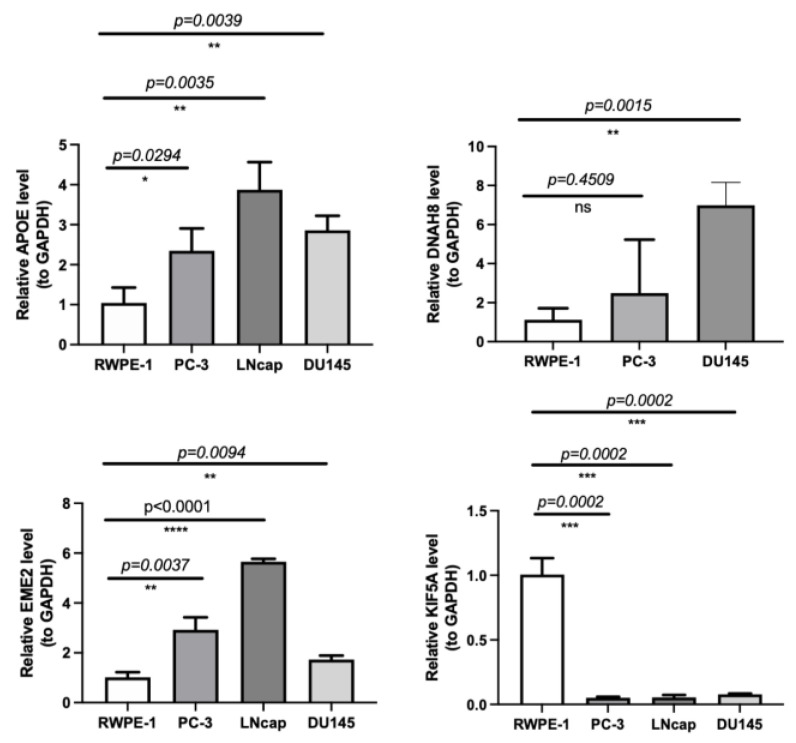
mRNA expression levels of the four model genes in human normal prostate epithelial cells (RWPE-1) and PCa cells (PC-3, LNcap, and DU145). * *p* < 0.05, ** *p* < 0.01, *** *p* < 0.001, **** *p* < 0.0001, ns represents no significant difference.

**Table 1 jcm-12-00654-t001:** Top 9 DE-MR-BCRGs by univariate Cox regression analysis.

Characteristics	HR *	HR.confint.lower	HR.confint.upper	*p*
APOE	1.56	1.27	1.91	0.000018
CDK1	1.52	1.23	1.88	0.000139767
CPT1B	1.89	1.48	2.41	0.000000381
CXCL8	0.91	0.78	1.06	0.226723867
CYP27A1	0.83	0.71	0.98	0.02487276
DNAH8	1.19	1.06	1.33	0.002902725
EME2	2.15	1.67	2.77	0.000000003
FOXH1	1.72	1.36	2.16	0.0000046
KIF5A	1.35	1.14	1.6	0.000651201
MTFR2	1.69	1.36	2.11	0.00000251
PAH	0.91	0.83	1.01	0.071111128

* HR—hazard ratio.

**Table 2 jcm-12-00654-t002:** Correlations between Clinicopathologic Features and the Risk Score in the Training Set.

		Risk Group		
Clinical Feature	Sample Number	High	Low	*p*-Value
PSA	386 (96.5%)			
≥10	10 (2.6%)	6	4	0.5216
<10	376 (97.4%)	187	189	
Race	387 (96.7%)			
White	333 (86.0%)	164	169	0.7277
Others	54 (24.0%)	28	26	
Gleason Score	400 (100%)			
≥7	361 (90.2%)	189	172	0.0041
<7	39 (9.8%)	11	28	
Age	400 (100%)			
≥60	233 (58.2%)	126	107	0.054
<60	167 (41.8%)	74	93	
Clinical stage	327 (81.7%)			
≥T3	39 (11.9%)	26	13	0.0279
<T3	288 (88.1%)	138	150	

**Table 3 jcm-12-00654-t003:** Correlations between Clinicopathologic Features and the Risk Score in the Validation Set.

		Risk Group		
Clinical Feature	Sample Number	High	Low	*p*-Value
PSA	248			
≥10	198 (79.8%)	110	88	0.9549
<10	50 (20.2%)	28	22	
Gleason Score	248			
≥7	206 (83.1%)	121	85	0.0299
<7	42 (16.9%)	17	25	
Age	248			
≥60	219 (49.6%)	123	96	0.651
<60	29 (50.4%)	15	14	
Clinical stage	223			
≥T3	96 (43.0%)	55	41	0.3648
<T3	127 (57.0%)	65	62	

**Table 4 jcm-12-00654-t004:** Results of independent prognostic univariate and multivariate Cox analyses.

Characteristics	HR *	HR.confint.lower	HR.confint.upper	*p*
Age	0.98	0.94	1.02	0.318994167
Gleason score	2.35	1.74	3.19	0.0000000299
T stage	1.92	0.93	3.98	0.079834582
APOE	1.43	1.12	1.82	0.004250533
DNAH8	1.28	1.14	1.45	0.0000718
EME2	2.4	1.8	3.21	0.00000000295
KIF5A	1.34	1.1	1.63	0.003934716
riskscore	1.27	1.2	1.34	0

* HR—hazard ratio.

**Table 5 jcm-12-00654-t005:** Correlations between the Functional Enrichment Results and the Risk Score.

Pathway	Correlation Coefficient	*p*-Value
KEGG adipocytokine signaling pathway	0.03029407	0.59219
KEGG alzheimers disease	0.13670246	0.01518
KEGG fatty acid metabolism	0.03029407	0.59219
KEGG ppar signaling pathway	−0.349770798	0.00000000017
KEGG primary bile acid biosynthesis	−0.384583119	0.00000000000152
GOBP cholesterol catabolic process	−0.318018926	0.00000000782
GOBP organic hydroxy compound biosynthetic process	−0.318018926	0.00000000782
GOBP small molecule catabolic process	−0.278421254	0.000000513
GOBP steroid catabolic process	−0.318018926	0.00000000782
GOMF cytoskeletal motor activity	0.087884393	0.11956
GOMF microtubule motor activity	0.087884393	0.11956

**Table 6 jcm-12-00654-t006:** qRT-PCR Results for the 4 model genes.

Gene	RWPE-1	PC-3	LNcap	DU145	f Value	*p* Value
APOE	1.0436 ± 0.3830	2.3466 ± 0.5615	3.8717 ± 0.6922	2.8600 ± 0.3614	15.5	0.0011
DNAH8	1.1212 ± 0.5931	2.4756 ± 2.7475	not detected out	6.9891 ± 1.1722	12.17	0.0024
EME2	1.0015 ± 0.2082	2.9219 ± 0.5027	5.6579 ± 0.1132	1.7273 ± 0.1614	149.5	<0.0001
KIF5A	1.0054 ± 0.1271	0.0519 ± 0.0077	0.0546 ± 0.0181	0.0787 ± 0.0058	161.3	<0.0001

## Data Availability

The TCGA-PRAD dataset was downloaded from the TCGA database (https://portal.gdc.cancer.gov/, accessed on 22 June 2022), GSE116918 was obtained from the GEO database (https://www.ncbi.nlm.nih.gov/gds/, accessed on 22 June 2022), and 785 MRGs were downloaded from GeneCards (https://www.genecards.org/, accessed on 24 June 2022).

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
