# Peer review of "A Novel Four Mitochondrial Respiration-Related Signature for Predicting Biochemical Recurrence of Prostate Cancer"

_jcm, 2023, doi:10.3390/jcm12020654_

Round 1

Reviewer 1 Report

Dear authors,

I have read your paper with great interest. In my opinion, the paper is well structured and, as you wrote in the discussion, the number of cases should be increased. But I think it's already a good start.

I only must ask you for some small clarifications that I report below.

-       The images in the figure 1, in my opinion, is too small. I ask you to reorganize it and specify in the text the tools you used to carry out the analysis.

-       The organization of tables 1, 2 and 3 needs to be reviewed in the text. You must go back and forth in the text, and you lose fluency in reading.

-       The image in the figure 3 is too small. I ask to the author it is possible to organize the figure such as figure 2.

-       In section 3.7, lane 359-360 DNAH8 expression is not reported for the LNCAP cell line. I assume it’s not significant, is it? Please, the authors to specify this in the text. If it is not significant, I would not include it in the graph and specify it in the figure legend. In table 6 I would write in the LNCAP column "not significant".

-       In the lane 467 I think that you must be added “Peroxisome proliferator-activated receptors (PPAR)” reported in the lane 469. 

Author Response

Reviewer 1:

Comment 1: The images in the figure 1, in my opinion, is too small. I ask you to reorganize it and specify in the text the tools you used to carry out the analysis.

Reply 1: We would like to thank the reviewer for this suggestion. We reorganized Figure 1. At the same time, we have added a 2.9 statistical analysis section to the manuscript to supplement the tools used in statistical analysis. Please see the revised manuscript on lines 209-215.

Comment 2: The organization of tables 1, 2 and 3 needs to be reviewed in the text. You must go back and forth in the text, and you lose fluency in reading.

Reply 2: Thanks for the advice. We made changes to the typesetting of Table1, Table2, and Table3. Please check it in the manuscript with tracking changes.

Comment 3: The image in the figure 3 is too small. I ask to the author it is possible to organize the figure such as figure 2.

Reply 3: We thank the reviewer for pointing out this question. We are sorry for our negligence. We have reorganized and updated Figure 3. Please check it in the manuscript with tracking changes.

 Comment 4: In section 3.7, lane 359-360 DNAH8 expression is not reported for the LNCAP cell line. I assume it’s not significant, is it? Please, the authors to specify this in the text. If it is not significant, I would not include it in the graph and specify it in the figure legend. In table 6 I would write in the LNCAP column "not significant".

Reply 4: We thank the reviewer for pointing out this question. We are sorry for our negligence. In the actual PCR, the fluorescence ct value of DNAH8 was not detected in LNcap cell line. According to the opinions of the reviewers, we updated Figure 6 and Table 6. Please check it in the revised manuscript.

Comment 5: In the lane 467 I think that you must be added “Peroxisome proliferator-activated receptors (PPAR)” reported in the lane 469. 

Reply 5: We thank the reviewer for pointing out this question. We have made corresponding changes in the revised manuscript. Please see lines 472-475 of the discussion section.

Reviewer 2 Report

The manuscript by Xia Z et al describes a novel signature for the prediction of BCR based on the expression of four mitochondrial respitaion genes. This is a very interesting paper and adds important knowledge for tackling PC recurrence and for disease management. I have minor comments that I think will make the paper more sound:

1- In the abstract (2nd line), it is mitochondrial respiration and not respiratory.

2- Is it possible to test the validity and robustness of the signature by applying it to different data sets representing different geographical/ethnical inputs? fo rexample, test it against a european cohort, USA cohort and Asia cohort.

3- Is it possible to validate the expression of the 4-gene signature by WB in the cells used (or at least couple of the genes)?

4- is it possible to validate the expression of the 4-gene signature by Immunohistochemical analyses on tissue sections from different stages of the disease? (again couple of genes would suffice).  

Author Response

Reviewer 2:

Comment 1: In the abstract (2nd line), it is mitochondrial respiration and not respiratory.

Reply 1: We thank the reviewer for pointing out this question. We are sorry for our mistake, we correct "mitochondrial respiratory" to "mitochondrial respiration".

Comment 2: Is it possible to test the validity and robustness of the signature by applying it to different data sets representing different geographical/ethnical inputs? fo rexample, test it against a european cohort, USA cohort and Asia cohort.

Reply 2: Thank you for the comments. According to the results of Table 2, there was no significant difference in high and low risk scores between Race (other whites) (pause 0.7277). In addition, we used two different regional data sets GSE70768 (UK) and GSE55945 (USA) data sets to verify the expression of APOE, DNAH8, EME2 and KIF5A. Our results found that the expression of APOE, DNAH8 and EME2 was up-regulated in normal samples, but there was no significant difference in the expression of KIF5A. Please see lines 354-359 in the discussion section.

Comment 3: Is it possible to validate the expression of the 4-gene signature by WB in the cells used (or at least couple of the genes)?

Reply 3: We thank the reviewer’s valuable suggestion. At present, due to funding constraints, we can not supplement the cell experimental verification. However, we added GSE70768 and GSE55945 data sets to verify the expression of the four model genes, and found that APOE, DNAH8, and EME2 were significantly lower expressed in PCa samples compared to normal samples, which is consistent with the results of qRT-PCR verification. Please see manuscript lines 364-372. Once we have obtained sufficient funding, we will verify the results of this study in follow-up studies.

Comment 4: Is it possible to validate the expression of the 4-gene signature by Immunohistochemical analyses on tissue sections from different stages of the disease? (again couple of genes would suffice).

Reply 4: We thank the reviewer’s valuable suggestion. At present, due to funding constraints, we can not supplement immunohistochemical analysis to verify the expression of four characteristic genes. However, we added GSE70768 and GSE55945 data sets to verify the expression of four model genes, and all found APOE, DNAH8, and EME2 were significantly lower expressed in PCa samples compared to normal samples, which is consistent with the results of qRT-PCR verification. Please see manuscript lines 364-372. Once we have obtained sufficient funding, we will verify the results of this study in follow-up studies.
